# Genome-Wide Identification and Analysis of the Heat-Shock Protein Gene in *L. edodes* and Expression Pattern Analysis under Heat Shock

Xu Zhao [1,2,3], Kaiyong Yin [1], Rencai Feng [1,2], Renyun Miao [1,2], Junbin Lin [1,2], Luping Cao [4], Yanqing Ni [5], Wensheng Li [5] and Qin Zhang [1,2,*]

[1]  Institute of Urban Agriculture, Chinese Academy of Agricultural Sciences, Chengdu 610299, China
[2]  Chengdu National Agricultural Science and Technology Center, Chengdu 610299, China
[3]  Facility Agriculture and Equipment Research Institute, Gansu Academy of Agri-engineering Technology, Wuwei 733006, China
[4]  College of Life Science and Technology, Gansu Agricultural University, Lanzhou 730070, China
[5]  College of Food and Biological Engineering, Chengdu University, Chengdu 610106, China
*  Correspondence: zhangqin@caas.cn

**Abstract:** *Lentinula edodes* (*L. edodes*), one of the most popular edible mushrooms in China, is adversely affected by high temperature. Heat shock proteins (HSPs) play a crucial role in regulating the defense responses against the abiotic stresses in *L. edodes*. Some HSPs in *L. edodes* have been described previously, but a genome-wide analysis of these proteins is still lacking. Here, the HSP genes across the entire genome of the *L. edodes* mushroom were identified. The 34 *LeHSP* genes were subsequently classified into six subfamilies according to their molecular weights and the phylogenetic analysis. Sequence analysis showed that LeHSP proteins from the same subfamily have conserved domains and one to five similar motifs. Except for Chr 5 and 9, 34 *LeHSPs* genes were distributed on the other eight chromosomes. Three pairs of paralogs were identified because of sequence alignment and were confirmed as arising from segmental duplication. In *LeHSPs*' promoters, different numbers of heat shock elements (HSEs) were predicted. The expression profiles of *LeHSPs* in 18N44 and 18 suggested that the thermo-tolerance of strain 18N44 might be related to high levels of *LeHSPs* transcript in response to heat stress. The quantitative real-time PCR (qRT-PCR) analysis of the 16 *LeHSP* genes in strains Le015 and Le027 verified their stress-inducible expression patterns under heat stress. Therefore, these comprehensive findings provide useful in-depth information on the evolution and function of *LeHSPs* and lay a theoretical foundation in breeding thermotolerant *L. edodes* varieties.

**Keywords:** *L. edodes*; heat shock protein; heat stress; genome analysis; RT-qPCR





## 1. Introduction

The basidiomycete *Lentinula edodes* (Berk.) Singer (Xianggu in Chinese or Shiitake in Japanese) is an edible mushroom that originates from China. It was first cultivated in China approximately 800 years ago and is currently the second most commonly produced edible mushroom in the world, with applications as decoctions, essences, and alternative medicines [1,2]. In 2020, the *L. edodes* production reached 11.64 million tons in China, accounting for 90% of the total global production (https://www.chyxx.com/industry/202201/992805.html (accessed on 10 Octorber 2022)).

During the cultivation of *L. edodes*, the environmental temperature is extremely rigorous and the optimum temperature for mycelia growth is 24~27 °C. Growth temperature higher than 32 °C will result in poor mycelia, and growth temperature higher than 35 °C can lead to severe retardation or even death. Hence, heat stress is considered one of the most important environmental stressors challenging *L. edodes* production.

To reduce the adverse effects under heat stress, fungus has evolved a series of self-defense mechanisms against heat shock to protect its body from damage [3,4]. Heat shock

proteins (HSPs), also referred to as heat stress proteins, are a kind of highly conserved proteins across primitive prokaryotes (bacteria) to higher level eukaryotes [5]. The first HSP was discovered in *Drosophila melanogaster* under heat stress [6]. Later, many members of HSP families were found in almost all living organisms. HSPs are produced in response to thermal stress as well as various stressful conditions including drought, salinity, extreme temperatures, chemical toxicity, and oxidative stress [7–9], thereby making HSPs extremely important "moonlight" protein families [10]. The expression of HSP genes is regulated by heat shock factors (HSFs). These HSFs can recognize and bind to specific DNA sequences such as 5′-nGAAC-3′ [11].

Members of HSP families share a highly conserved domain named "heat-shock" and are classified into six major categories according to their molecular weights and sequence homologies: HSP100, HSP90, HSP70, HSP60, HSP40, and small HSP (SHSP) [12].The cellular mechanisms underlying the function of HSPs under abiotic stress are not fully understood, but many of them were found to act as chaperones or co-chaperones involved in protein synthesis and folding [7], in the degradation of abnormal protein [13], and in cellular localization [14] during stress conditions. HSPs in a few model plants have been well characterized. *Arabidopsis* 21, 27, 18, 7, and 8 genes have been identified for encoding, including the HSF, SHSP, HSP70, HSP90, and HSP100 families, respectively [15]. Genome-wide identifications of HSPs have been performed in wheat, soybean, and rice [12,16–18]. To date, limited information is available on genome-wide identification of the entire HSP families in *L. edodes*. The response of HSPs to heat stress is unclear and clarifying the mechanism of heat stress tolerance in *L. edodes* is vital.

HSP families play a crucial role in response to abiotic stress, but the relationship between heat stress resistance and HSP expression is ambiguous in *L. edodes*. Therefore, a total of 34 *HSP* genes are identified in the *L. edodes* genome with bioinformatics methods. Then, the physicochemical characteristics, chromosomal location, phylogenetic tree, gene structure, conserved motifs, homologous gene pairs, expression in response to heat stress, and heat shock elements (HSEs) in the promoter of *LeHSP* genes are analyzed. The *LeHSP* expression profiles under heat stress indicate that different combinations of *LeHSPs* are involved in the thermotolerance of strains 18N44 and Le027. This study systematically elucidates the evolution of HSPs in *L. edodes* and relevant role of *LeHSP* genes in response to heat stress. Thus, this work not only lays a foundation for further research on the biological function of *LeHSP* genes, but also provides a reference base for the molecular mechanism of heat resistance breeding in *L. edodes*.

## 2. Materials and methods

### 2.1. Identification of HSPs in L. edodes

The protein sequences of *L. edodes* were downloaded from the NCBI database (https://www.ncbi.nlm.nih.gov/genome/?term=lentinula+edodes (accessed on 1 July 2022)) and used for the construction of the local database [19]. To identify the HSP family, the hidden Markov models of the HSP20 (PF00011), HSP40 (PF00226 and PF01556), HSP60 (PF00118), HSP70 (PF00012), HSP90 (PF00183), and HSP100 domains (PF02861 and PF10431) were downloaded from the Pfam database (https://pfam.xfam.org/ (accessed on 4 July 2022)) [20] and used as queries ($p < 0.0001$) to search against the local database of *L. edodes* by using the Simple HMMER search plug-in implemented in the TBtools software (v1.098773) [21]. To ascertain all HSP family members, a blastp search was performed by employing all HSP amino acid sequences from the HMMER search as queries against the non-redundant protein sequences of *L. edodes* with an e-value of $1e^{-4}$. After the removal of the redundant HSP proteins, the putative HSP proteins were verified for the presence of HSP related domains using other databases including the NCBICDD (https://www.ncbi.nlm.nih.gov/cdd (accessed on 18 September 2022)), HAMMER scan (https://www.ebi.ac.uk/Tools/hmmer/search/hmmscan (accessed on 18 September 2022)), and SMART (http://smart.embl-heidelberg.de/ (accessed on 18 September 2022)). The identified genes were named with a prefix 'Le' for HSP40, HSP60, HSP70, HSP90, HSP100 and 'LeS' for

small HSP, and numbered in increasing order with their position on the chromosome proceeding from the short to the long arm.

## 2.2. Chromosomal Localization and Gene Structure

The chromosomal localizations and intron–exon structures of *HSP* genes were retrieved from the annotation files of *L. edodes* and visualized by Map Genes on Genome and Gene Structure View (Advanced), respectively. Both plug-ins were implemented in the TBtools software. The motifs of HSP proteins were identified by the MEME program (Version 5.4.1, https://meme-suite.org/meme/tools/meme (accessed on 4 July 2022)) [22] with the parameter of a maximum of 5 motifs and other parameters as default. The number of amino acids, molecular weights, and theoretical isoelectric points (pI) were analyzed by the Geneious primer software (https://www.geneious.com/prime/ (accessed on 6 May 2022)) [23].

## 2.3. Duplication of LeHSPs in L. edodes

Genes were defined as paralogues if they meet the following criteria: (a) the length of the alignable sequence covers more than 75% of the longer gene; and (b) the aligned regions have more than 75% similarity. In addition, paralogous pairs were defined as tandemly duplicated genes if they are separated by five or fewer genes within a 100-kb region; if the paralogous pairs were mapped onto duplicated chromosomal fragment, then they were designated as segmental duplications [24].

## 2.4. Phylogenetic Reconstruction

The full amino acid sequences of the HSPs of members from *Agaricusbisporus*, *Pleurotuspulmonarius*, *Pleurotuseryngii*, and *Volvariellavolvacea* were downloaded from the NCBI.

Phylogenetic analysis was performed using the Geneious primer software (Version 2022.2). In brief, all LeHSP protein sequences were first aligned according to MUSCLE Alignment with default parameters in Geneious primer. A Geneious tree was subsequently constructed based on the aligned sequences using the UPGMA method with the Jukes–Cantor model. The final phylogenetic tree was embellished and visualized by iTOL (https://itol.embl.de/ (accessed on 1 July 2022)) [25] and Adobe Illustrator 2020.

## 2.5. Heat Shock Element Analysis in the Promoter Regions of HSP Genes

The promoter sequences (1 kb upstream of the initiation ATG start codon) of each *HSP* gene were extracted using the Gtf/Gff3 Sequences Extrator plug-in TBtools. The locations and numbers of the heat stock element (HSE, 5′nGAAC-3′) were predicted and visualized by TBtools.

## 2.6. Strain Materials and High-Temperature Treatments

*L. edodes* strains 18 and 18N44 were used for RNA-seq analysis. Both strains were obtained from the Institute of Edible Fungi, Shanghai Academy of Agricultural Sciences. Strain 18 is an elite *L. edodes* in agronomic traits but is sensitive to heat stress, and strain 18N44 was selected from the protoplasts of the original strain 18 which was exposed to ultra-violet rays. Strain 18N44 showed good performance in both agronomic traits and thermotolerance [26]. A similar method described by Zhao et al. [27] with some modification was used for high-temperature treatment. In brief, the mycelia of both strains 18 and 18N44 were incubated at 25 °C for 14 days on PDA medium and then cultured in the PDB medium. After incubating at 25 °C for 14 days with shaking at 150 r/min, the mycelia were treated with high-temperature stress at 37 °C for 0, 4, 12, and 24 h. The mycelia were collected and immediately frozen in liquid nitrogen and stored at −80 °C for RNA isolation and RNA-seq. Three biological replications were carried out for each sample. Two other *L. edodes* strains Le015 and Le027 were used for expression analysis. Strain Le015 is sensitive to heat-stress, and Le027 is a thermo-resistant strain. The same cultivation conditions and heat stress treatments were applied for strains Le015 and Le027.

The mycelia were collected for RNA isolation, and the obtained RNAs were used for qPCR analysis.

### 2.7. Extraction and Purification of Total RNA from Mycelia

Total RNA was isolated using the Fungal Total RNA Isolation Kit (B518629, Sangon Biotech, China) and purified following the manufacturer's protocol. The purity and concentration of RNA was measured by NanoDrop 2000 C (Thermo Fisher Scientific, Waltham, MA, USA).

### 2.8. RNA-Sequencing and Analysis of HSP Genes

The qualified RNA samples from strains 18 and 18N44 were sent to Oebiotech (Shanghai, China) for transcriptome sequencing. The FPKM [28] and read counts value of each unigene was calculated using bowtie2 [29] and eXpress [30]. The heatmap of the gene expression pattern was generated according to the FPKM values using the R (version 4.0.4) package heatmap (1.0.8).

### 2.9. Gene Expression by RT-qPCR

Quantitative reverse transcription-PCR was carried out using II Green One-Step qRT-PCR SuperMix (TransScrip, China) according to the manufacturer's instructions. The reference genes of *TUB* and *UBI* [27,31,32] were used to normalize the differences between samples. The relative expressions of *HSP* genes were calculated using the $E^{\Delta\Delta Ct}$ method [33] against the geometric mean of two internal reference genes. The primers were designed by Primer 5.0. The primers for RT-qPCR are listed in Table S1. The results were calculated by Excel and displayed as means $\pm$ standard deviation (SD).

## 3. Results

### 3.1. Identification and Analysis of HSP Genes in L. edodes

A total of 34 *HSP* genes were identified in the *L. edodes* genome, including eight small *HSP* (*SHSP*), six *HSP40*, nine *HSP60*, six *HSP70*, two *HSP90*, and three *HSP100* genes (Table 1). These 34 *HSP* genes were further named based on their chromosomal locations. The protein sequence lengths of LeHSPs varied from 153 to 890 amino acids with molecular weights ranging from 17.41 kDa to 98.436 kDa. The theoretical isoelectric point (pI) values of the LeHSPs ranged from 4.4 (HSP70.6) to 10.3 (LeSHSP.1).

**Table 1.** Molecular features of *Hsp* genes in *L. edodes*.

|  | Name | Protein ID | Exon | Intron | Location | Chr | AA | MW(kDa) | PI |
|---|---|---|---|---|---|---|---|---|---|
| HSP100 | LeHSP100.1 | XP_046091847.1 | 9 | 8 | 6279738–6282864 | 1 | 890 | 98.436 | 5.48 |
|  | LeHSP100.2 | XP_046082612.1 | 11 | 10 | 2522518–2526440 | 2 | 771 | 84.959 | 5.59 |
|  | LeHSP100.3 | XP_046080071.1 | 11 | 10 | 3743879–3748058 | 6 | 749 | 83.108 | 5.54 |
| HSP90 | LeHSP90.1 | XP_046088585.1 | 8 | 7 | 2407716–2410082 | 1 | 706 | 80.185 | 4.62 |
|  | LeHSP90.2 | XP_046090891.1 | 19 | 18 | 1815650–1819056 | 5 | 817 | 91.712 | 4.29 |
| HSP70 | LeHSP70.1 | XP_046088664.1 | 5 | 4 | 2657999–2660171 | 1 | 651 | 70.962 | 4.78 |
|  | LeHSP70.2 | XP_046088714.1 | 9 | 8 | 2800610–2802917 | 1 | 613 | 67 | 5.24 |
|  | LeHSP70.3 | XP_046086623.1 | 11 | 10 | 4172952–4175420 | 3 | 627 | 68.046 | 5.46 |
|  | LeHSP70.4 | XP_046083696.1 | 9 | 8 | 1532467–1535557 | 8 | 886 | 97.597 | 5.52 |
|  | LeHSP70.5 | XP_046079422.1 | 9 | 8 | 1824230–1826698 | 8 | 678 | 73.742 | 4.74 |
|  | LeHSP70.6 | XP_046090274.1 | 4 | 3 | 288215–290346 | 10 | 631 | 66.324 | 4.41 |
| HSP60 | LeHSP60.1 | XP_046082828.1 | 9 | 8 | 1726567–1728602 | 1 | 539 | 57.88 | 7.33 |
|  | LeHSP60.2 | XP_046088531.1 | 8 | 7 | 2303108–2305148 | 1 | 554 | 59.722 | 6.24 |
|  | LeHSP60.3 | XP_046092210.1 | 6 | 5 | 4944807–4946672 | 1 | 560 | 60.881 | 5.98 |
|  | LeHSP60.4 | XP_046084390.1 | 9 | 8 | 1792382–1794487 | 4 | 550 | 59.833 | 6.63 |
|  | LeHSP60.5 | XP_046084477.1 | 4 | 3 | 1298157–1300114 | 7 | 598 | 62.592 | 5.26 |
|  | LeHSP60.6 | XP_046084950.1 | 8 | 7 | 1721053–1723408 | 7 | 575 | 62.377 | 5.79 |
|  | LeHSP60.7 | XP_046088311.1 | 11 | 10 | 2236637–2238719 | 7 | 542 | 59.243 | 5.29 |
|  | LeHSP60.8 | XP_046083662.1 | 8 | 7 | 1633717–1635690 | 8 | 561 | 60.138 | 5.91 |
|  | LeHSP60.9 | XP_046079390.1 | 10 | 9 | 1750215–1752273 | 8 | 524 | 56.462 | 5.16 |

**Table 1.** *Cont.*

|  | Name | Protein ID | Exon | Intron | Location | Chr | AA | MW(kDa) | PI |
|---|---|---|---|---|---|---|---|---|---|
| HSP40 | LeHSP40.1 | XP_046088819.1 | 6 | 5 | 3097898–3099296 | 1 | 398 | 43.547 | 6.44 |
|  | LeHSP40.2 | XP_046091483.1 | 6 | 5 | 7116980–7118740 | 1 | 493 | 52.394 | 9.53 |
|  | LeHSP40.3 | XP_046084554.1 | 11 | 10 | 1053764–1055625 | 7 | 422 | 46.417 | 4.59 |
|  | LeHSP40.4 | XP_046084878.1 | 12 | 11 | 1936586–1938308 | 7 | 368 | 41.501 | 6.18 |
|  | LeHSP40.5 | XP_046079519.1 | 4 | 3 | 2621252–2622535 | 7 | 372 | 39.857 | 9.66 |
|  | LeHSP40.6 | XP_046082726.1 | 5 | 4 | 2888022–2889557 | 7 | 429 | 48.317 | 10.02 |
| SHSP | LeSHSP.1 | XP_046080632.1 | 4 | 3 | 3183573–3184343 | 2 | 190 | 21.261 | 10.33 |
|  | LeSHSP.2 | XP_046088441.1 | 3 | 2 | 5037923–5037741 | 3 | 154 | 17.671 | 6.54 |
|  | LeSHSP.3 | XP_046084164.1 | 2 | 1 | 1241615–1242240 | 4 | 188 | 20.37 | 4.83 |
|  | LeSHSP.4 | XP_046088091.1 | 3 | 2 | 2943783–2944370 | 6 | 156 | 17.666 | 5.65 |
|  | LeSHSP.5 | XP_046088092.1 | 3 | 2 | 2944818–2945400 | 6 | 155 | 17.409 | 5.45 |
|  | LeSHSP.6 | XP_046082167.1 | 3 | 2 | 382364–382938 | 7 | 153 | 17.491 | 5.57 |
|  | LeSHSP.7 | XP_046082168.1 | 4 | 3 | 383366–383941 | 7 | 162 | 18.492 | 6.45 |
|  | LeSHSP.8 | XP_046080378.1 | 3 | 2 | 2094217–2095081 | 7 | 254 | 29.054 | 5.22 |

*3.2. Gene Structure of HSPs*

The intron–exon structures were determined by the alignment of the genomic DNA sequences with the coding sequences of *HSPs* (Figure 1). The number of exons of *LeHSP* genes varied from 2 to 19 exons (Figure 1, Table 1). *LeSHSP* genes had 2 to 4 exons. The number of exons in *LeHSP40* genes ranged from 4 to 12, with 3 in *LeHSP40.5* and 11 introns in *LeHSP40.4*. The number of exons of *LeHSP60s* were between 4 (*LeHSP60.5*) and 11 (*LeHSP50.7*). The exons number in *LeHSP70s* ranged from 4 to 11. Two *LeHSP90* genes were identified from *L. edodes*. *LeHSP90.1* had 8 exons, and *LeHSP90.2* had 19. Three *LeHSP100* genes were identified, within two of them (*LeHSP100.2* and *LeHSP100.3*) had 11 exons, and the remainder (*LeHSP100.1*) had only 8 exons.

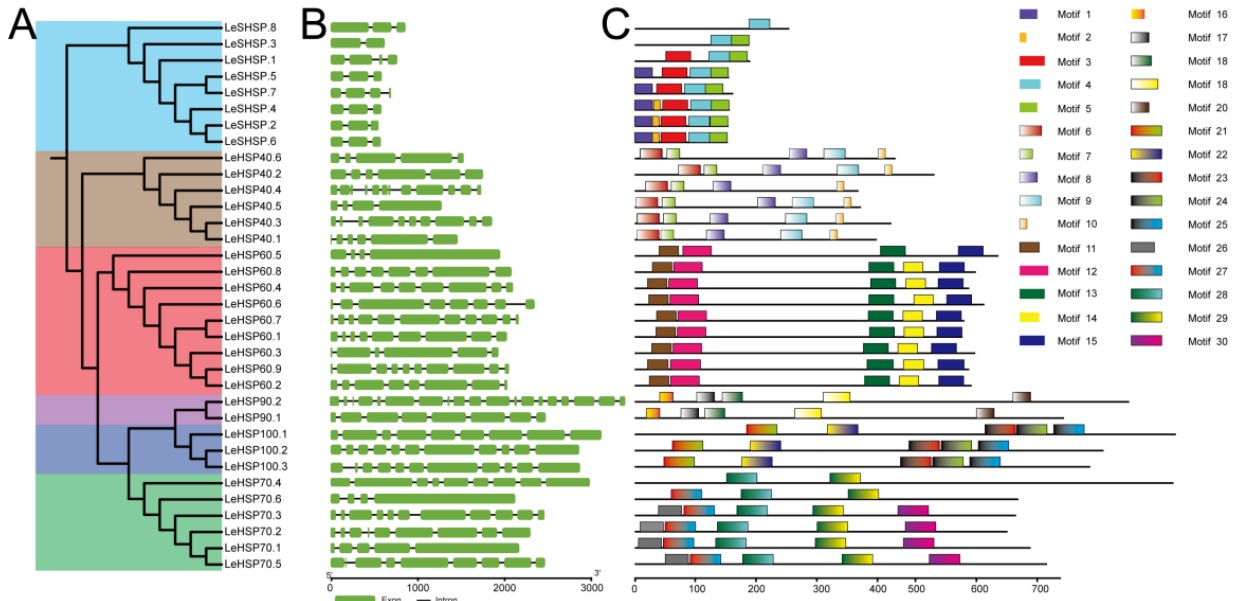

**Figure 1.** Phylogenetic relationship, gene structure, and conserved motif analysis of *LeHSP* genes. (**A**) Phylogenetic tree of 34 LeHSP proteins. Azure, brown, red, purple, blue, and green colors represent the *LeSHSP*, *LeHSP40*, *LeHSP60*, *LeHSP90*, *LeHSP100*, and *LeHSP70* genes, respectively. (**B**) Exon/intron organization of *LeHSP* genes. Green boxes represent exons and black lines with the same length represent introns. (**C**) Distributions of conserved motifs in LeHSPs. The conserved motifs of 34 proteins were identified and visualized by MEME.

Domain analysis can help clarify the functions of the HSPs. The conserved domains of LeSHSP, LeSHP40, LeHSP60, LeHSP70, LeHSP90, and LeHSP100 were identified by Bath Web CD-Search tool (https://www.ncbi.nlm.nih.gov/Structure/cdd/wrpsb.cgi (accessed on 18 September 2022)) against the CDD, Pfam, and SMART databases.

LeSHSP.1, LeSHSP.3, and LeSHSP.8 had a conserved ACD domain consisting of 80 to 100 amino acids (Table S2). The other LeSHSPs contained the conserved IbpA domain (cl00175). Two conserved domains, DnaJ_C (PF1556) and DnaJ (PF00226), existed in all LeHSP40 proteins. LeHSP60s contained a compound domain Cpn60_TCP1 (PF00118). Two LeHSP70s (LeHSP70.4 and LeHSP70.6) had a conserved domain HSP70 superfamily (cl37948). The other four LeHSP70s contained another HSP70 (PF00012) domain. LeHSP90.1 and LeHSP90.2 contained two conserved domains, HSP90 (PF000183) and ATPase_c. LeHSP100s contained four conserved domains including AAA_2 (PF077724), AAA_lid_9 (PF17871), AAA (PF00004), and ClpB_D2_small (PF10431). In addition to the four conserved domains mentioned above, LeHSP100.1 had another specific domain, Clp_N (PF02861).

The sequence motifs of each LeHSP member were individually predicted by the MEME tool (Table S3). The maximum number of motifs for each protein was set to five. The sequence motifs were conserved within but not between subfamilies, as evidenced by the absence of common motifs in all HSPs (Figure 1). The number of conserved motifs in each HSP subfamily also varied. For instance, of the five motifs predicted in members of LeSHSPs, only one motif was shared by all LeSHSP members. By contrast, all five predicted motifs in the LeHSP100 subfamily were shared by each member.

Except for LeHSP40.4 which possessed four conserved motifs, the other LeHSP40s contained five conserved motifs. LeHSP60s had five conserved motifs, except for LeHSP60.5 which lacked Motif 14. LeHSP70.4 only had two conserved motifs which were Motifs 28 and 29. Compared to LeHSP70.4, LeHSP70.6 had extra Motif 27. Five conserved motifs were found in LeHSP70.1, LeHSP70.2, LeHSP70.3, and LeHSP70.5. LeHSP90s and LHSP100s contained five conserved motifs. Detailed information on motifs is presented in Table S3.

### 3.3. Phylogenetic Analysis of HSP Proteins

The full-length amino acid sequences of HSP proteins from *L. edodes*, *A. bisporus*, *P. pulmonarius*, *P. eryngii*, and *V. volvaceas* were used for phylogenetic tree construction (Figure 2). The 186 HSP sequences were categorized into three distinct groups according to the sequence characteristics. Group I was subdivided into three subgroups representing members from SHSPs, HSP90s, and HSP100s. The SHSP subfamily was an extensive subgroup which included 41 protein sequences. Group II consisted of 43 HSP70 sequences. All members from the HSP40 and HSP60 subfamilies were clustered into Group III. Within Group III, all HSP60 sequences and one HSP40 sequence (Accession number: KAF8652705) from *V. volvaceas* were clustered into a subgroup (Subgroup I). The other 31 HSP40 sequences were clustered into a separate subgroup (Subgroup II).

### 3.4. Chromosomal Location and Duplication of LeHSP Genes

Chromosomal location analysis revealed that the 34 *LeHSP* genes were unevenly distributed on all chromosomes of *L. edodes* with the exemption of Chromosomes 5 and 9 (Figure 3A). Chromosome 7 contained the largest number (10) of *LeHSPs* followed by Chromosome 1 with nine genes. Only one gene was positioned on Chromosome 10 (*LeHSP70.6*). Chromosomes 6 and 8 contained four *LeHSP* genes. Chromosomes 2, 3, and 4 each possessed two *LeHSP* genes. No rule was discerned for the distribution of different types of *LeHSP* genes. Eight *LeSHSP* genes were, respectively, located on Chromosomes 2 (1 gene), 3 (1 gene), 4 (1 gene), 6 (2 genes), and 7 (3 genes). Six *LeHSP40* genes were distributed on Chromosomes 1 (2 genes) and 7 (4 genes). Chromosomes 1, 4, 7, and 8 contained three, one, three, and two *LeHSP60* genes, respectively. Six *LeHSP70* genes were mapped on Chromosomes 1 (2 genes), 3 (1 gene), 8 (2 genes), and 10 (1 gene). Two *LeHSP90* genes were located on Chromosomes 1 and 6. Three *LeHSP100* genes were scattered on Chromosomes 1, 2, and 6.

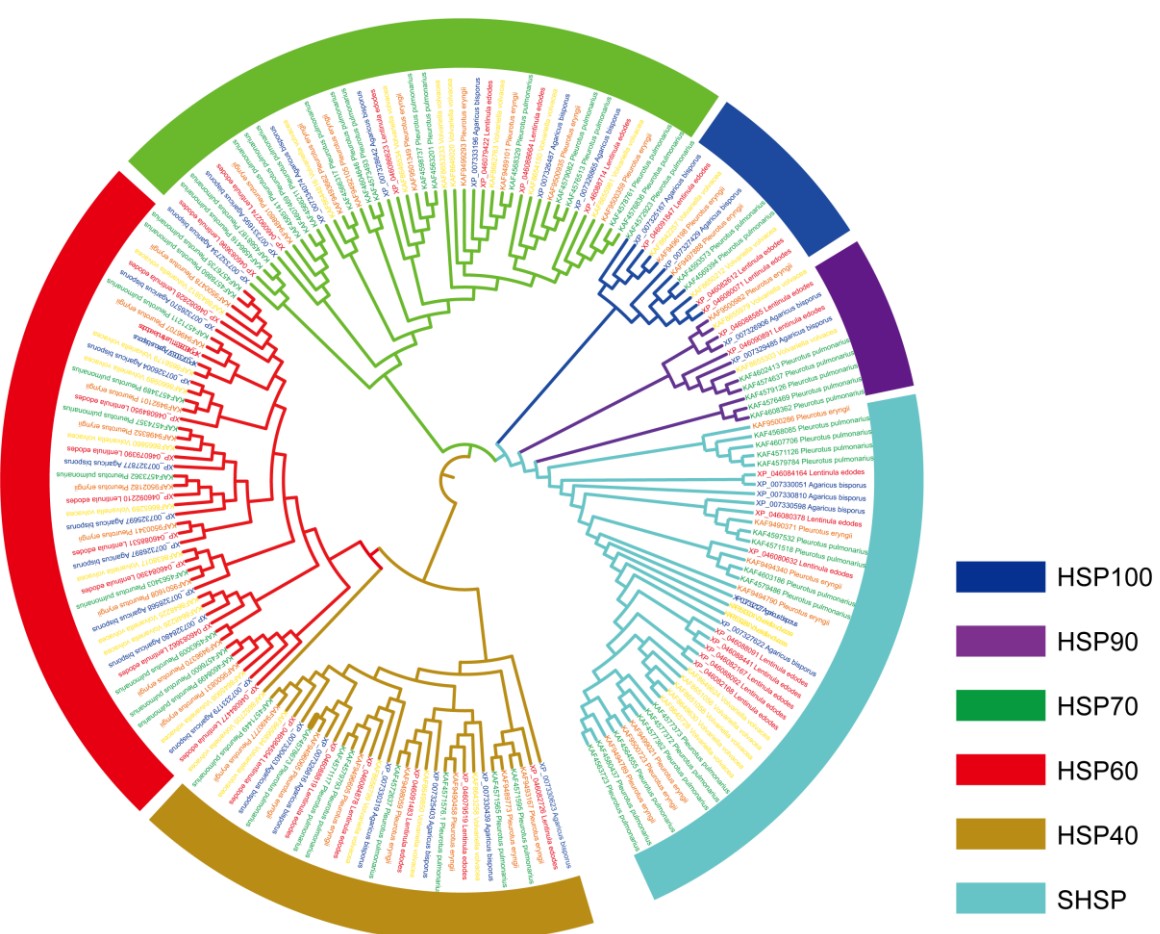

**Figure 2.** Phylogenetic tree of HSPs from *L. edodes, A. bisporus, P. pulmonarius, P. eryngii*, and *V. volvaceas.* The phylogenetic tree was constructed using the UPGMA method and Jukes-Cantor model with the Geneious primer software. Different subfamilies are shaded with different colors.

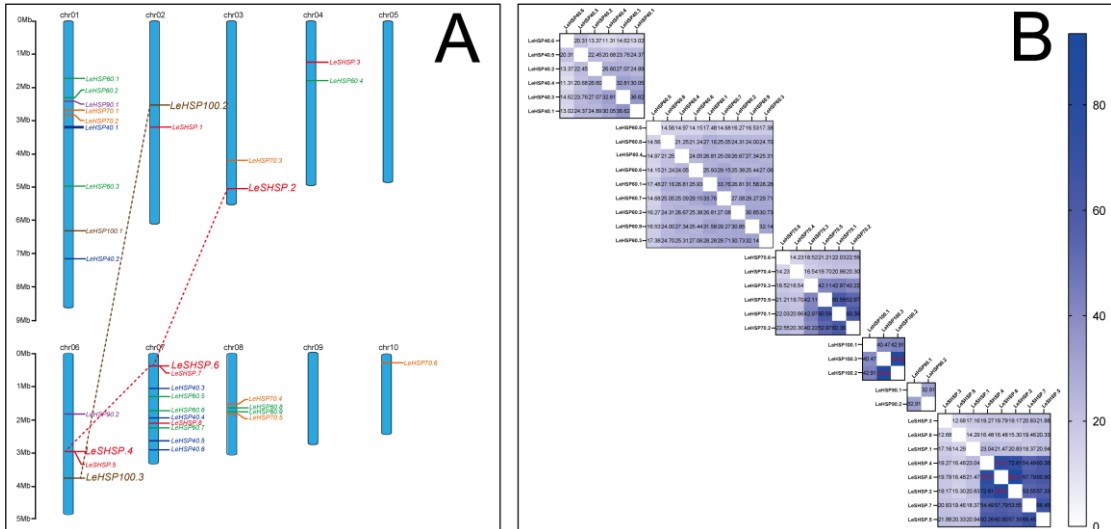

**Figure 3.** Chromosomal location and gene duplication of *LeHSP* genes. (**A**) 34 *LeHSP* genes were mapped on 8 chromosomes. Chromosome numbers are indicated at the top of each bar. The gene names are labelled in different colors. The scale on the left is in megabases. (**B**) Relationship of *LeHSPs* with one another in different subfamilies. The white color represents shared zero percent identity, and deep blue indicates shared 100% identity.

One the basis of the evolutionary relationships among the coding sequences of the 34 *LeHSP* genes, one and two paralogous pairs were observed in *LeHSP100* and *LeSHSP* genes, respectively (Figure 3B). All pairs were segmental duplications according to Rudd et al. [34].

### 3.5. The Binding Site of HSF in the Promoters of LeHSP Genes

The HSE 5′-nGAAC-3′ was specifically recognized by the HSF, and the affinity between HSF and HSE was affected by the number of HSEs displayed on the promoter. To further analyze the combining capacity between *LeHSF* and the promoter of *LeHSP*, the 1 kb upstream sequences from the initiation codon (ATG) of *LeHSP* genes were extracted and the number of HSEs was identified (Figure 4). All *LeHSP* genes' promoters possessed at least one HSE, suggesting that HSEs might play pivotal roles in response to heat stress. However, the number of HSEs in individual genes highly varied. For instance, the promoters of *LeHSP60.2* contained only one HSE, and the promoters of *LeSHSP.2* had up to nine HSEs.

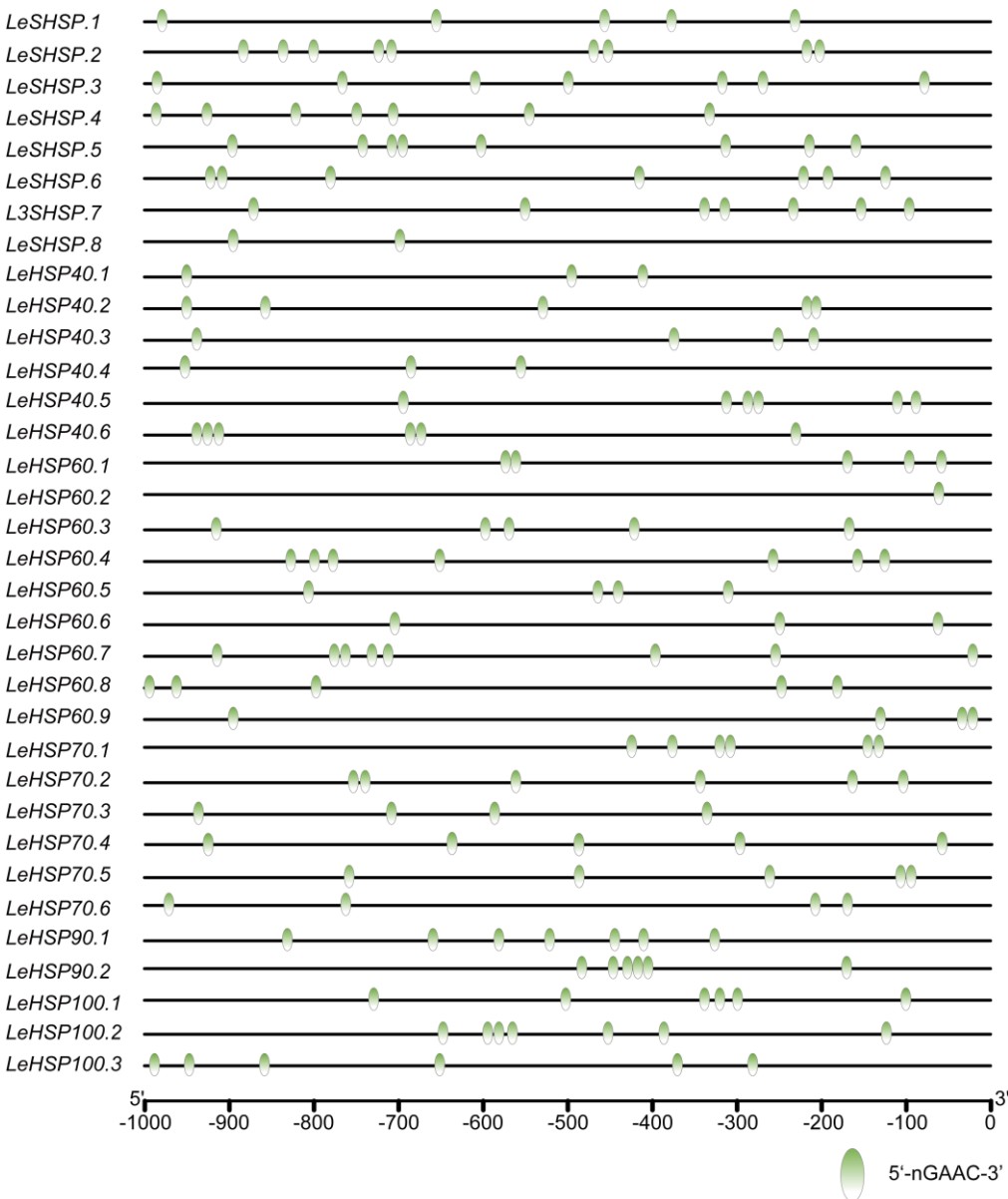

**Figure 4.** Predicted binding site of HSF in *LeHSP* promoters. Promoter sequences (−1000 bp) of 34 *LeHSP* genes were used to predict the HSEs (5−nGAAC−3′) using TBtools. The location of HSEs can be inferred according to the scale at the bottom.

### 3.6. Expression Profile of LeHSP Genes in Response to Heat Stress

To understand the function of *LeHSP* genes in response to heat stress, the expression patterns of LeHSP genes in both Strains 18 and 18N44 (thermo-resistant) under heat-stress were investigated (Figure 5). However, the FPKM values of the other four *LeHSP* genes (LeSHSP.6, LeSHSP.8, LeHSP40.3, and LeHSP40.6) were zero or undetectable in the RNA-seq data, so this part of the expression data was discarded during the analysis. Thirty *LeHSP* genes had FPKM values and were up-expressed in response to heat stress. These *LeHSP* genes were divided into three groups according to the resulting expression pattern. In Group I, 12 *LeHSP* genes were highly expressed at 4 or 12 h under heat stress in Strain 18. In Group II, 11 *LeHSP* genes were clustered and showed a high level of expression at 24 h in Strain 18N44. In Group III, 6 of 7 *LeHSP* genes were highly expressed at 24 h in Strains 18 and 18N44. However, the expression pattern of LeHSP.4 was unusual as it was highly expressed at 24 h in Strain 18 and low expressed at 24 h in Strain 18N44.

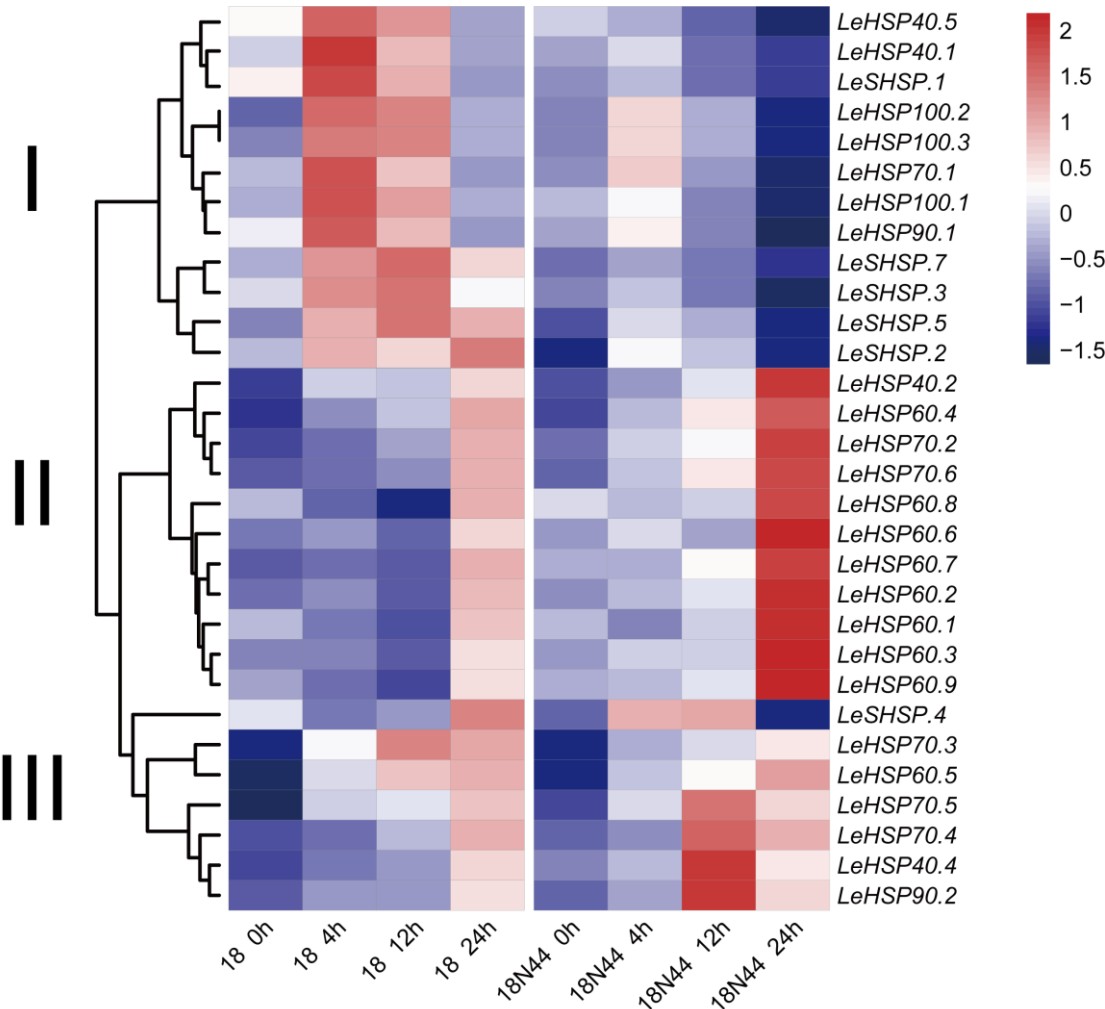

**Figure 5.** Heat map of the expression profiles of *LeHSP* genes in response to heat stress. The FPKM values of *LeHSP* genes were Z-scores normalized and centralized by a unit variance scaling method.

To confirm the expressions of *LeHSPs* in response to heat stress, RT-qPCR was performed at the same time-points of heat stress in two other strains (Le015 and Le027) by selecting 16 *LeHSP* genes which had representative expression profiles (Figure 6). According to the RT-qPCR results, most *LeHSPs* were sensitive to heat stress and showed different expression patterns during the 24 h heat stress. The expressions of *LeSHSP.1*, *LeSHSP.4*, and *LeSHSP.5* were up-regulated under heat stress and peaked at 4 h. The relative expressions of *LeSHSP.4* and *LeSHSP.5* increased dramatically by more than 400-fold under

heat stress compared to the control (without heat stress treatment) in both Le015 and Le027. An increase in expression was observed for the *LeHSP40* genes (*LeHSP40.1*, *LeHSP40.2*, and *LeHSP40.4*) at 4 h of heat stress, and *LeHSP40.1* and *LeHSP40.2* peaked after 4 h of heat stress. The expressions of *LeHSP60s* (*LeHSP60.3*, *LeHSP60.4*, and *LeHSP60.6*) were down-regulated in response to heat stress. The expression of *LeHSP70.2* was unchanged under heat stress. *LeHSP70.3* was up-regulated only after 4 h of heat stress, peaked at 4 h (Le027) or 12 h (Le015), and then dropped subsequently. The expression of *LeHSP70.4* was increased after 12 h of heat stress and maintained a high level after that period of heat stress. *LeHSP90s* were extremely induced by heat stress in Strain Le027, but the expression of *LeHSP90.2* in Strain Le015 was repressed under heat stress. The expression levels of *LeHSP100s* were up-regulated and peaked at 4 h of heat shock, then dropped subsequently. Compared to Le015, the expression of LeHSP100 was significantly higher in Le027.

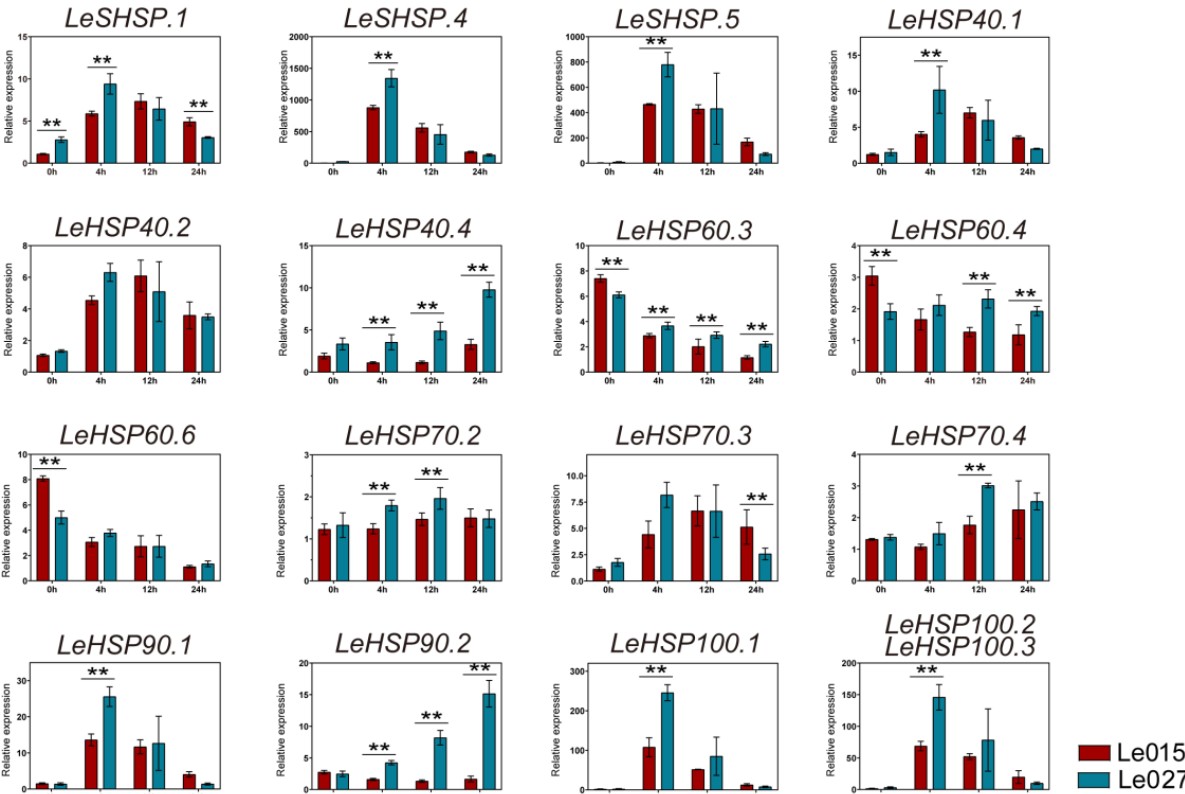

**Figure 6.** Expression profiles of *LeHSP* genes under heat stress in the strains of Le015 and Le027. Transcript level of different *LeHSP* genes (relative to geometric mean of *LeTUB* and *LeUBI*) in response to heat stress in Le015 (red) and Le027 (blue). X-axes showed heat stress time. Y-axes indicated scales of relative expression level. One way ANOVA was used for statistical analysis. *p*-values are indicated as follows ** $p < 0.01$.

## 4. Discussion

HSPs play a crucial role in prokaryotes and eukaryotes in response to various stress conditions, including heat stress [35]. To date, the identifications and functions of *HSPs* have been well investigated in animals and plants but largely lagged in edible fungi. In the present study, 34 HSP genes, with 8 *LeSHSPs*, 6 *LeHSP40s*, 9 *LeHSP60s*, 2 *LeHSP90s*, and 3 *LeHSP100s*, were identified from the entire genome of *L. edodes* (Le(Bin) 0899 ss11). Previous research [36] identified 11 *LeSHSPs*, 1 *LeHSP60*, 13 *LeHSP70s*, 2 *LeHSP90s*, 1 *LeHSP98* and 1 *LeHSP104* in Strains YS606 and YS3357 of *L. edodes* through RNA-seq data. We believed that two reasons may be responsible for the different number of *LeHSPs* in the two studies. Firstly, the variation in the number of HSPs found in *L. edodes* may be caused by the different genome sources used in these two studies. Secondly, some *LeHSP* genes

expressed under the limit of detection using RNA sequencing and these genes were not analyzed in Wang's study.

Compared to other species, *L. edodes* had fewer *HSP* genes. A total of 64, 79, and 753 HSP genes were identified in *A. thaliana* [16], *S. italica* [37], and *T. aestivum* [12], respectively. The low number of HSP genes in *L. edodes* may arise from its small genome. For instance, *A. thaliana*, whose genome size is 125 Mb [38], possesses few *HSP* genes, but *T. aestivum* with 6.3 Gb of genome size [12] has numerous HSP genes.

In this study, we identified 34 *HSP* genes in *L. edodes*, then analyzed the structure, chromosomal location, gene duplication, binding site of HSF, and expression pattern under heat stress. Variation occurred in terms of the number of introns and exons, but the motifs were conserved in different classifications of HSP proteins. The phylogenetic and motif analysis of HSPs in *L. edodes* confirm that members from the same sub-group share the same features of conserved domains, indicating they possess a similar biological function in response to different stresses. The different numbers of HSEs laid in the *LeHSPs'* promoter revealed that all *LeHSP* genes have the potential to be recognized by HSFs [39]. Gene duplications play a key role in expansions of gene family members in many species, such as tomato [40], pepper [41], and cucumber [42]. In this study, a total of three gene duplications were found from the *L.edodes* genome and these duplications belonged to segmental duplication events, implying that the three duplication events contributed to the expansion of the *LeHSP* family.

According to the expression patterns, most *LeHSPs* were expressed in response to heat stress. In Strains 18 and 18N44, the transcript level of all *LeHSP* genes showed an increase after 4 h to 24 h of heat stress at 37 °C (Figure 5). In the thermo-tolerant Strain 18N44, the expression level of *LeHSPs*, including *LeHSP40.2*, *LeHSP60s*, *LeHSP70.2* and *LeHSP70.6*, were significantly up-regulated under heat stress. In previous study, they confirmed that the relative expression of *LeHSP100*, *LeHSP90*, and *LeHSP60* is far higher in 18N44 than that in strain 18 [43]. The RT-qPCR results reveal that except for *LeHSP60* (*LeHSP60.3*, *LeHSP60.4*, and *LeHSP60.6*) whose expression showed a decrease in response to heat stress, the expression levels of the 13 other *LeHSPs* (*LeSHSP*, *LeHSP40*, *LeHSP70*, *LeHSP90* and *LeHSP100*) were higher under heat stress than under the normal condition (25 °C) (Figure 6). Given the results, we deduced that the combination of different types and numbers of up-regulated *HSP* expression can contribute to thermotolerance. Previous studies have successfully examined the function of HSP40 (LeDnaJ07) in *L. edodes* and found that overexpression of *LeDanJ07* could enhance mycelial resistance in response to heat stress in YS55 [2]. Identification and expression of DnaJ protein (HSP40) was well studied in *L. edodes*, and the expression of *LeDnaJ* genes was induced by cadmium, Trichoderma atroviride, and heat stress, thereby indicating that LeDnaJ proteins were involved in multiple stresses [44]. In addition, the proteome analysis exhibited that the contents of HSP40 (DnaJ), HSP70, and HSP98 were significantly up-regulated in the thermotolerant Strain S606 after heat stress [36]. Up-regulation of members of various *HSP* genes in response to heat stress are reported in other species [45,46]. For example, 2-fold to 20-fold increases in expressions were detected for 11 members of the *A. thaliana* HSP70 family under heat stress [16]. In finger millet, a HSP gene, HSP17.8, showed a 40-fold up-regulation under heat stress, was cloned, and proved to have major contributions to thermo-tolerance in the crop [47]. The results also confirmed the association of *HSP* genes (HSP70, HSP21, and SHSP21) with the high temperature tolerance of *Agasicles hygrophila* [48]. The expression of *HSP* genes were regulated by the HSFs. When cells were exposed to abiotic stress, HSFs promptly bind to the promoter of *HSPs* to regulate gene transcription. Then, the newly synthesized HSPs act as molecular chaperone to prevent the substrate protein from aggregating irreversibly [49,50]. Subsequently, the different signaling pathways related to enhancing tolerance to heat stress will be activated by HSPs. In this study, we confirmed that thermotolerance was associated with up-regulated HSP expression, but the mechanism by which HSPs enhance thermo-resistance in *L. edodes* requires further investigation.

## 5. Conclusions

In this study, 34 *HSPs* were identified and characterized after scanning for the *L. edodes* genome. LeHSPs were classified according to their molecular weight and verified by phylogenetic tree construction and conserved motifs analysis. The gene structure, distribution, motif analysis, duplication, and phylogenetic analysis revealed a clear evolutionary history for the HSP family in *L. edodes*. Three gene duplication events were found in the *LeHSP* gene family, and they belong to segmental duplications. The different numbers of HSEs in the *LeHSPs'* promoters indicated that HSEs could be induced by heat stress.

The expression analysis indicated that the combination of different types and numbers of up-regulated *LeHsp* genes mainly contributed to thermo-tolerance. This study not only lays a scientific foundation for in-depth understanding of the regulation mechanism of HSP genes, but also provides a novel direction for breeding a mycelium of *L. edodes* that is tolerant to high temperature.

**Supplementary Materials:** The following supporting information can be downloaded at: https://www.mdpi.com/article/10.3390/cimb45010041/s1, Table S1. Primer pairs used for gene expression. Table S2. Information of Domains in HSPs. Table S3. Amino sequence of motifs in HSPs.

**Author Contributions:** X.Z. and Q.Z. were involved in the design and coordination of the study, conceptualized the experiments, analyzed data, and produced the manuscript. K.Y., R.F. and R.M. participated in the design and coordination of the study, conceived of the experiments, analyzed data, and contributed to manuscript writing. J.L. conceived of experiments and contributed to manuscript writing. L.C., Y.N. and W.L. performed experiments and analyzed data related to the RT-qPCR. All authors have read and agreed to the published version of the manuscript.

**Funding:** This work was supported by the Scientific and Technological Innovation Talents of Sichuan Province (Grant No. 2022JDRC0034), the Gansu Provincial Youth Science and Technology Fund Program (Grant No. 20JR5RA064), the Central Public-interest Scientific Institution Basal Research Fund (Grant No. S2022007), and the Local Financial Funds of the National Agricultural Science and Technology Center, Chengdu (Grant No. NASC2021KR06).

**Institutional Review Board Statement:** Not applicable.

**Informed Consent Statement:** Not applicable.

**Acknowledgments:** The authors acknowledge Xiaojiang Guo for his input in bioinformatics analysis.

**Conflicts of Interest:** The authors declare no conflict of interest.

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
