# Peer review of "Genome-Wide Identification and Analysis of the Heat-Shock Protein Gene in L. edodes and Expression Pattern Analysis under Heat Shock"

_cimb, doi:10.3390/cimb45010041_

Round 1

Reviewer 1 Report

In this manuscript, Zhan et al. a whole genome sequencing was performed in L.edodes. A total of 34 HSPs were identified. Their expression profiles were investigated together with a comprehensive study of the phylogenetic relationship, gene structure and conserved domains. Overall, the study provides sights on understanding regulation mechanism of HSP genes. A meticulous and professional language editing throughout the text is highly recommended before being published. 

Author Response

Dear Review,

Thanks your affirmation for my manuscript. As you mentioned the language, we invited English polishing agency to edit this manuscript.

Qin

Reviewer 2 Report

Thank you for the opportunity to review this manuscript, dealing with interesting findings entitled “Genome-Wide Identification and Analysis of the heat-Shock 2 Protein Gene in L.edodes and Expression pattern Analysis un- 3 der Heat shock s”. All findings are interesting, and the article includes a balanced and critical view of the outcomes. Following comments can improve the quality of articles before it

1.      In my opinion, the research background should be improved.  In order to balance the scientific viewpoint and attract more attention from audiences, the authors are highly recommended to consider the inclusion of the some recent relevant research and review papers 

2.      Hypothesis of study is missing. Incorporate hypothesis in  introduction section

3.      Most of methods are without any citation. include citation

4.      The quality of Figure 6 is poor need to improve. Furthermore the error bars must be indicated by letter based on LSD value 

5.      Most of the reference are too old. At least 50 % reference should be from last five years

6.      Conclusions need to be revised as per own results

7.       The whole article needs to be checked critically for typos and grammar errors

8.      All the references should be according to Journal guidelines

Author Response

Dear Editors and Reviewers,

Thank you for your letter and for the reviewers’ comments concerning our manuscript entitled “Genome-Wide Identification and Analysis of the Heat-Shock Protein Gene in L. edodes and Expression Pattern Analysis under Heat Shock” (cimb-2099158). Those comments are all valuable and very helpful for revising and improving our paper, as well as the important guiding significance to our research. We have carefully concerned the suggestion of reviewer and make some changes. Revised portion are marked in red in the paper. The main corrections in the paper and the responds to the reviewer’s comments are as flowing:

Comment 1: In my opinion, the research background should be improved.  In order to balance the scientific viewpoint and attract more attention from audiences, the authors are highly recommended to consider the inclusion of the some recent relevant research and review papers.

Response 1: I have carefully modified the introduction and add some new references in this part. As you mentioned some references were too old, this is a truth. For some viewpoint or finding we have cited, we should traced the source.

Comment 2: Hypothesis of study is missing. Incorporate hypothesis in introduction section.

Response 2: In the last paragraph of introduction, we have proposed hypothesis which is HSP families play a crucial role in response to abiotic stress, but the relationship between heat stress resistance and HSP expression is ambiguous in L. edodes in line 81-82.

Comments 3: Most of methods are without any citation. include citation.

Response 3: Citations were added in some methods or software. (Seen in line 101, 103, 120, 124, 141,)

Comments 4: The quality of Figure 6 is poor need to improve. Furthermore the error bars must be indicated by letter based on LSD value.

Response 4: Figure 6 was a tif format. When type setting, I can offer the EPG format pictures. As you mentioned the error bars must be indicated by letter based on LSD value, I used SPSS software to analyze their significant difference, see Figure 6. I will upload a tif format in CIMB.

Comment 5: Most of the reference are too old. At least 50 % reference should be from last five years.

Response 5: I have added some references from last five years. Seen in the submission.

Comment 6: Conclusions need to be revised as per own results

Response 6: We have rewrite the conclusions. (See in line 387-390)

Comment 7: The whole article needs to be checked critically for typos and grammar errors

Response 7: We sent this paper to English polishing agency for polishing. See in resubmission.

Comment 8: All the references should be according to Journal guidelines

Response 8: All reference style was used according to MDPI in Endnote.

Reviewer 3 Report

This is excellent work leading to next practical application. However, what was flagged in the conclusion (lines93-94) is a very long and difficult pathway.

Author Response

Dear Editors and Reviewers,

Thank you for your letter and for the reviewers’ comments concerning our manuscript entitled “Genome-Wide Identification and Analysis of the Heat-Shock Protein Gene in L. edodes and Expression Pattern Analysis under Heat Shock” (cimb-2099158).

As you mentioned in line 93-94, there is not conclusion. Could you please give me comments in detail.

Qin

Reviewer 4 Report

The authors of the manuscript, ‘CRBIOT-D-22-00188’ performed a Bioinformatic analysis of four Eucalyptus tree species. Specifically, quantifying the number of COR genes present, investigating their conserved domains and motifs, presence of the cis-regulatory elements, gene structure and arrangements, and their phylogenetic relationships. Like any other gene Bioinformatics research, this study is novel and sheds light in the field of Biotechnology. However, I believe this study can be enhanced, as it is it does not provide any new insights. Listed below are some of my comments:

Major Comments

1.      I suggest the authors to perform a comparative Bioinformatic analysis of these four plant species. In the present study, the work does not exhaust all the Bioinformatic information that can be gleaned.

2.    The authors used much of the Present Simple tense in result description. Better to revise the whole manuscript and use the correct tense especially in result description.

3.   The MS should be improved with more analyses like the Synteny relations, and Gene duplication analyses.

4.      In the Phylogenetic analysis, a combined phylogenetic tree including all the species in study may be included.

Minor Comments

1.      Polish English usage and proficiency.

2.     The result section should separated into different subsections to avoid confusion and improve clarity. 

Author Response

Dear Editors and Reviewers,

Thank you for your letter and for the reviewers’ comments concerning our manuscript entitled “Genome-Wide Identification and Analysis of the Heat-Shock Protein Gene in L. edodes and Expression Pattern Analysis under Heat Shock” (cimb-2099158). Those comments are all valuable and very helpful for revising and improving our paper, as well as the important guiding significance to our research. We have carefully concerned the suggestion of reviewer and make some changes. Revised portion are marked in red in the paper. The main corrections in the paper and the responds to the reviewer’s comments are as flowing:

Major Comments

Comment 1: I suggest the authors to perform a comparative Bioinformatic analysis of these four plant species. In the present study, the work does not exhaust all the Bioinformatic information that can be gleaned.

Response 1: We want to analyze the heat shock protein in Lentinula edodes. We have analyzed gene structure, chromosome location, conserved motifs, phylogenetic analysis and genes expression in response to heat stress. For phylogenetic analysis, we analyzed the relationship in L. edodes, A. bisporus, P. pulmonarius, P. eryngii, and V. volvaceas.

Comment 2: The authors used much of the Present Simple tense in result description. Better to revise the whole manuscript and use the correct tense especially in result description.

Response 2: In the results parts, we used past indefinite tense to describe the results. We invited English polishing agency to edit this manuscript.

Comment 3: The MS should be improved with more analyses like the Synteny relations, and Gene duplication analyses.

Response 3: Thanks for your comments, it is very useful for me. The gene duplication was present in the results. We just used a method that used in previous study: Genes were defined as paralogues if they meet the following criteria including: (a) length of the alignable sequence covers more than 75% of the longer gene; (b) similarity of the aligned regions are more than 75%.

Comment 4: In the Phylogenetic analysis, a combined phylogenetic tree including all the species in study may be included.

Response 4: In this paper, the Hsp gene sequences were used from edible fungus, like Agaricus bisporus, Pleurotus pulmonarius, Pleurotus eryngii and Volvariella volvacea. We want to know whether hsps from different sub-families were conserved.

Minor Comments

Comment 5: Polish English usage and proficiency.

Response 5: We invited English polishing agency to edit this manuscript.

Comment 6: The result section should separated into different subsections to avoid confusion and improveclarity. 

Response 6: In the results parts, we used six subtitles to show our results.

If you have any questions about my response, please do not hesitate to contact me.

Qin

Reviewer 5 Report

The manuscript is easily readable. Nevertheless, I have spotted a number of sentences that could be improved in grammar or structure. Also, punctuation and typography should be checked (e.g., latin letters for genes and species). Furthermore, please check the proper explanation of abbreviations and spell out species names when using them for the first time.

The title should be corrected: e.g., plural: genes; upper and lower case letters; punctuation, and maybe even revised. For the reasons please consider my remarks below.

The article provides informative data, however, its message is unclear to me. For this reason, I think that some of the information is not necessarily needed to support the message. But again, the message is still unclear, so it is hard for me to advise.

My general concern is that the data is not presented in a convincing way, mostly because it is not clear whether the focus is on the identification of HSPs or rather on the expression patterns and potential functions. I find it confusing, and as the reader I would not like to look for details and bring them together, this should be done by the authors in the first place.

In several cases, it is not clear why the authors decided to use specific strains or species, or methods of calculation. I would like to know more about the rationale, for instance, concerning the construction of the phylogenetic tree and the results. As mentioned above, I am also not sure whether the presentation of the phylogenetic tree really adds substantial information to the manuscript (despite that it is for sure relevant per se).

I also do not fully agree with the presentation of the heat shock response. On one hand a heat map is presented that in my understanding was created using two strains, one thermo-resistant and one thermo-sensitive. The expression was then, according to the authors, confirmed by RT-qPCR using different strains. However, based on the current presentation the two experimental set-ups are not linked in a way so that I can understand how successful this was.

Concerning the presentation of gene structure, it would maybe be useful to select specific genes based on their heat stress response and then concentrate the presentation of the results on these. For example, it is not very convincing if the authors write that “domain analysis can help to better understand the functions of the HSPs” but then do not reflect on this in the following.

Indeed, the discussion tries to focus on these aspects but again these are not linked to the overall results. The discussion takes out some examples but does not bring them into context. In my opinion, again the missing focus of the manuscript is reflected here, as the discussion seems to be a random collection.

I feel that the conclusions give some guidance on the messages of the manuscript. This may serve as a starting point to refine the message.

The references need to be revised and unified in their presentation. I also advise to check for the relevance of the cited references when the message of the manuscript has been clarified.

Author Response

Dear Editors and Reviewers,

Thank you for your letter and for the reviewers’ comments concerning our manuscript entitled “Genome-Wide Identification and Analysis of the Heat-Shock Protein Gene in L. edodes and Expression Pattern Analysis under Heat Shock” (cimb-2099158). Those comments are all valuable and very helpful for revising and improving our paper, as well as the important guiding significance to our research. We have carefully concerned the suggestion of reviewer and make some changes. Revised portion are marked in red in the paper. The main corrections in the paper and the responds to the reviewer’s comments are as flowing:

The manuscript is easily readable. Nevertheless, I have spotted a number of sentences that could be improved in grammar or structure. Also, punctuation and typography should be checked (e.g., latin letters for genes and species). Furthermore, please check the proper explanation of abbreviations and spell out species names when using them for the first time.

Response:We invited English polishing agency to edit this manuscript.

The title should be corrected: e.g., plural: genes; upper and lower case letters; punctuation, and maybe even revised. For the reasons please consider my remarks below.

The article provides informative data, however, its message is unclear to me. For this reason, I think that some of the information is not necessarily needed to support the message. But again, the message is still unclear, so it is hard for me to advise.

My general concern is that the data is not presented in a convincing way, mostly because it is not clear whether the focus is on the identification of HSPs or rather on the expression patterns and potential functions. I find it confusing, and as the reader I would not like to look for details and bring them together, this should be done by the authors in the first place.

In several cases, it is not clear why the authors decided to use specific strains or species, or methods of calculation. I would like to know more about the rationale, for instance, concerning the construction of the phylogenetic tree and the results. As mentioned above, I am also not sure whether the presentation of the phylogenetic tree really adds substantial information to the manuscript (despite that it is for sure relevant per se).

Response: Concerning the method of construction of the phylogenetic tree, we used Geneious primer based on the aligned sequences using the UPGMA method with the Jukes-Cantor model. This is not a new tool or method. In other papers, they used this software to do sequence alignment.

I also do not fully agree with the presentation of the heat shock response. On one hand a heat map is presented that in my understanding was created using two strains, one thermo-resistant and one thermo-sensitive. The expression was then, according to the authors, confirmed by RT-qPCR using different strains. However, based on the current presentation the two experimental set-ups are not linked in a way so that I can understand how successful this was.

Response: I understand your doubt. A heat map we used two strains to present our RNA-seq data. For expression, we want to know whether these genes are still response to heat stress in other tolerant strains. This is not a special example, in other papers for gene family analysis, authors download RNA-se data online, and gene expression verification used their own material in lab was okay. In this manuscript, we have used two different strain to explain the expression pattern is reasonable.

Concerning the presentation of gene structure, it would maybe be useful to select specific genes based on their heat stress response and then concentrate the presentation of the results on these. For example, it is not very convincing if the authors write that “domain analysis can help to better understand the functions of the HSPs” but then do not reflect on this in the following.

Response: In this part, we mentioned that “domain analysis can help to better understand the functions of the HSPs”, then we found that different Hsp families had one or two conserved domain. In this paragraph, we want tell readers different sub-family of heat shock proteins have specific domains and these domains were reported previously have different functions.

Indeed, the discussion tries to focus on these aspects but again these are not linked to the overall results. The discussion takes out some examples but does not bring them into context. In my opinion, again the missing focus of the manuscript is reflected here, as the discussion seems to be a random collection.

Response: In discussion, we have improved our writing and discussed the results. We have rewrite the gene expression part in discussion. (See in line 342-375)

I feel that the conclusions give some guidance on the messages of the manuscript. This may serve as a starting point to refine the message.

Response: We have rewrite this party. (See in line 377-389)

The references need to be revised and unified in their presentation. I also advise to check for the relevance of the cited references when the message of the manuscript has been clarified.

Response: We have modified the style into MDPI style using Endnote software.

Round 2

Reviewer 4 Report

I recommend to accept it for publication.

Author Response

Hi editors and reviewer: 

Thanks for recognition of my work.

Reviewer 5 Report

Firstly, I would like to acknowledge the significant improvement of your manuscript and your responses to my remarks.

Please allow me a few sentences of clarification: The authors have answered by explaining the method of constructing the phylogenetic tree. However, that has not been the subject of my concern. Despite that my concern concerning the message of the manuscript has not been directly addressed by the authors, in the revised version it is clearer that the study concentrates on the identification of HSPs and not on their function. Similarly, I have not doubted the experimental setup of investigating the heat shock response but rather how the results were presented. Nevertheless, as stated above the presentation has improved to a sufficient extent.

Please consider that the discussion could still be improved by providing a clear link between your results and the cited research by adding your detailed findings before discussing the other studies (following the lines 340-342, which seem to me too general given your findings). E.g, you have found significant upregulation of some genes, which you may bring forward in the discussion.

Please note in the following some detailed comments that I propose for your consideration:

Line 110: “increasing position”? Please reconsider.

Line 194: Space between sentences.

Lines 196-198: “Three LeHSP100 genes…”: Please revise this sentence to make it less confusing. At least add: all of them, and remove: “all of them” (which should be two) and “the remainder” (which should be one).

Lines 191-192: “LeSHSP genes had 2 to 4 exons because of their short coding sequence.” It can be but is not necessarily a consequence. I suggest to reconsider this sentence.

Line 193: Please add “the number of exons”

Line 215: Please remove the brackets.

Line 222: Check spaces, comma.

Line 308: It might be useful to repeat here which strains you have used.

Line 309: Please remove “s” in 1 LeHSP60s.

Lines 310-312: I agree this might have been one of the reasons; as the differences are considerable could you think about any other potential reasons? Also, I suggest to check the order of the listed genes.

Line 318: Do you mean “genes” or indeed “gene families”?

Line 326: “all the potential they possessed” – the meaning is not clear to me.

Line 327: Should it be “contributed to the abundance”?

Lines 328-331: You may reconsider this sentence and simplify it.

Line 346: Where appropriate, change to italic and remove spaces.

Line 348: Please check whether italic letters are appropriate here.

Line 353: Up-regulation?

Line 357: “by the upstream” – please clarify.

Line 372: Should be “promoters”?

Line 394: “their” – correct?

Table S3: Add “amino acid sequence”

Author Response

Dear Editors and Reviewers,

Thank you again for your letter and for the reviewers’ comments concerning our manuscript entitled “Genome-Wide Identification and Analysis of the Heat-Shock Protein Gene in L. edodes and Expression Pattern Analysis under Heat Shock” (cimb-2099158). Those new comments are all valuable and very helpful for revising and improving our paper, as well as the important guiding significance to our research. We have carefully concerned the suggestion of reviewer and make some changes. Revised portion are marked in red in the paper. The main corrections in the paper and the responds to the reviewer’s comments are as flowing:

Please consider that the discussion could still be improved by providing a clear link between your results and the cited research by adding your detailed findings before discussing the other studies (following the lines 340-342, which seem to me too general given your findings). E.g, you have found significant upregulation of some genes, which you may bring forward in the discussion.

Response: In the discussion, we put forward an argument according our results: the combination of different types and numbers of up-regulated HSP expression can contribute to thermo-tolerance. In the following discussion, we provide several examples to explain the different combination of expression of Hsps can improve heat tolerance.

Please note in the following some detailed comments that I propose for your consideration:

Line 110: “increasing position”? Please reconsider.

Response:I deleted the “increasing” word, this sentence changed into numbered in increasing order with their position on the chromosome proceeding from the short to the long arm. See in line 110.

Line 194: Space between sentences.

Response:Done.

Lines 196-198: “Three LeHSP100 genes…”: Please revise this sentence to make it less confusing. At least add: all of them, and remove: “all of them” (which should be two) and “the remainder” (which should be one).

Response: I have rewritten this sentence. See in line 197-198.

Lines 191-192: “LeSHSP genes had 2 to 4 exons because of their short coding sequence.” It can be but is not necessarily a consequence. I suggest to reconsider this sentence.

Response: I have deleted the second half of this sentence. You are right, the second half should not be put in results parts. See in line 191-192.

Line 193: Please add “the number of exons”

Response: Thank you so much. I have done. See in line 193.

Line 215: Please remove the brackets.

Response: Done. See in line 215.

Line 222: Check spaces, comma.

Response:Done. See in line 222.

Line 308: It might be useful to repeat here which strains you have used.

Response: Hi reviewers and editors, the genome data was used which was downloaded from NCBI. I have added the strain which I used. See in line 308.

Line 309: Please remove “s” in 1 LeHSP60s.

Response:Done. See in line 309.

Lines 310-312: I agree this might have been one of the reasons; as the differences are considerable could you think about any other potential reasons? Also, I suggest to check the order of the listed genes.

Response: Done. See in line 309-315.

Line 318: Do you mean “genes” or indeed “gene families”?

Response: I have deleted “families”. See in line 321.

Line 326: “all the potential they possessed” – the meaning is not clear to me.

Response: I have rewritten this sentence. See in line 329-330.

Line 327: Should it be “contributed to the abundance”?

Response: I have rewritten this sentence. See in line 330-331.

Lines 328-331: You may reconsider this sentence and simplify it.

Resonse: I rewrote this sentence. See in line 332-334.

Line 346: Where appropriate, change to italic and remove spaces.

Response: Done. See in line 351.

Line 348: Please check whether italic letters are appropriate here.

Response: Done. See in line 353.

Line 353: Up-regulation?

Response: Done.See in line 358.

Line 357: “by the upstream” – please clarify.

Response: I want to explain that HSF was in upstream of HSP. If ambiguous, I have deleted “upstream of”. See in line 362.

Line 372: Should be “promoters”?

Response: Done. See in line 377.

Line 394: “their” – correct?

Response: Done. See in line 399.

Table S3: Add “amino acid sequence”

Response: Done. See in Table S3.

If this manuscript has some problems, please do not hesitate to contact me.

Qin